# Energy Efficient Wireless Signal Detection: A Revisit through the Lens of Approximate Computing



Abhinav Kulkarni *, Messaoud Ahmed Ouameur and Daniel Massicotte *

Electrical and Computer Engineering Department, Université du Québec à Trois-Rivières, Trois-Rivières, QC G9A 5H7, Canada; messaoud.ahmed.ouameur@uqtr.ca
* Correspondence: abhinav.kulkarni@uqtr.ca (A.K.); daniel.massicotte@uqtr.ca (D.M.)

**Abstract:** In the pursuit of energy efficiency in next-generation communication systems, approximate computing is emerging as a promising technique. In the proposed work, efforts are made to address the challenge of bridging the gap between the level of approximation and the Quality-of-Service (QoS) of the system. The application of approximate multiplication to wireless signal detection is explored systematically, illustrated by employing Truncated Multiplication (TM) on Quadrature Phase Shift Keying (QPSK) Minimum Mean Square Error (MMSE) detection. The irregularities induced by approximation in the multiplication operation employed in wireless signal detection are captured by the Approximate Multiplication Noise (AMN) model, which aids in the analysis of signal fidelity and resiliency of the system. The energy efficiency gains through approximation are highlighted in the approximation analysis. Signal fidelity analysis provides the capability to predict system output for varying levels of approximation, which aids in improving the stability of the system. The higher approximation levels are advantageous in low Signal-to-Noise Ratio (SNR) regimes, whereas lower approximation levels prove beneficial in high SNR regimes.

**Keywords:** wireless signal detection; approximate computing; energy efficiency; arithmetic multiplication; noise; resiliency

## 1. Introduction.

The explosive growth of mobile traffic, driven by the emergence of new services and applications [1] is propelling the development of next-generation communication systems. This surge in mobile traffic necessitates an increase in the communication system capability [2]. Crucial enabling technologies for next-generation communication systems operate at the spectrum-level, protocol-level, and infrastructure-level [3]. On the spectrum level, efforts are concentrated on increasing carrier frequencies, while at the protocol level, gains are projected through adjustments in data packet packaging. Infrastructure-level advancements in hardware technology play a pivotal role in enabling next-generation communication systems, where energy consumption is a key design factor impacting the scalability of the system [4].

The work [5] emphasizes the increasing importance of energy efficiency in system design, aligning it with considerations of spectral efficiency and spatial reuse. In wireless communication systems, energy consumption at Base Station (BS) and core networks became a noteworthy concern due to the imperative for extensive coverage, heightened BS density in populated areas, and management of data. Remarkably, a substantial portion, ranging from 60% to 80%, of the overall energy consumption is attributed solely to BS [6]. This energy consumption in active mode at the physical layer of BS is linked to baseband processing, RF processing, and signal power amplification. The average baseband energy consumption for a 4G LTE and 5G NR BS is approximately 150 and 220 Watts, respectively, constituting up to 5–15% of the total BS energy consumption [7]. Moreover, the energy efficiency of RF and power amplification is influenced by the efficiency of baseband processing [8].

The work [9] underscores the importance of integrating energy-efficient techniques into the design of baseband processing, addressing both operational costs and environmental concerns related to BS operation. Enhancing the energy efficiency of baseband processing, which plays a crucial role in BS operation, has a cascading effect on the overall energy consumption of the BS. Likewise, since wireless signal detection is fundamental to baseband signal processing, enhancing the energy efficiency of the former leads to an improvement in the energy efficiency of the latter. Going further, a system-wide perspective needs to include a real-time assessment and control of the energy consumption for wireless signal detection to cater to the incorporation of future technologies.

The following contributions are presented in this work:

- Modeling of a constant noise model for wireless signal detection to evaluate the impact of irregularities caused by approximate multiplication. AMN is a novel constant noise model that effectively captures irregularities of TM for QPSK MMSE signal detection.
- Gauging the effect of TM on Symbol Error Rate (SER) of QPSK MMSE signal detection. The derived analytical expression computes SER by using AMN.
- Proposition of resiliency metrics to provide insights into resilient TM configurations for QPSK MMSE signal detection. A TM configuration characterized by a low level of approximation proves advantageous in high SNR regimes, whereas one with a high level of approximation is preferable in low SNR regimes.

In Section 1, the motivation for the work is established, and the contributions of the proposed work are laid out. Section 2 outlines the related work. In Section 3, the methodology is detailed, utilizing components to derive entities and primary metrics for the proposed work. Section 4 presents the derivation of secondary metrics and analysis. Finally, in Section 5, the conclusion of the proposed work is presented.

## 2. Related Work

QOS in wireless communication encompasses a set of components that collectively define and manage the level of service quality. Bit Error Rate (BER), SER, and Frame Error Rate (FER) are key factors in ensuring the QOS, directly impacting received signal fidelity [10]. Monitoring and controlling these factors is crucial for maintaining the reliability and stability of communication systems, especially in accuracy-sensitive applications. Several efforts are made in integrating energy efficiency with QOS. Energy awareness has been induced at the algorithmic level for Internet-of-Things (IOT) application and analyzed as a QOS factor in the work [11]. The work [12] highlights the integrated approach of QOS parameters including energy efficiency and their influence on the dynamic network condition and mobility of wireless sensor nodes. The work [13] explores the development of an optimal radio resource allocation method in 5G LTE networks based on adaptive selection of channel bandwidth depending on the QOS requirements.

As there has been a recent outburst to employ Artificial Intelligence (AI) techniques for intelligent automation, this approach has also been explored to achieve energy efficiency in communication systems [14,15]. However, the direct application of this approach at the physical layer of communication systems can incur additional computational overheads [16] related to ancillary data processing, which could potentially negate the energy efficiency benefits obtained at the foremost. Moreover, baseband processing of the physical layer is also being incorporated into resource-constrained edge computing devices, where the power budget is very stringent, but the performance constraints might be relaxed. The work [17] underscores the challenges of meeting the commercial budget requirements of power consumption of communication systems designed for THz frequency, which limits their operating frequency.

Approximate computing intentionally introduces errors into systems to enhance energy and resource consumption efficiency, albeit at the expense of tunable accuracy loss [18,19]. This methodology relies on the error-resilient nature of applications and employs disciplined methods for inserting errors into the system. The approach spans hardware, software, and cross-layer methodologies across diverse application domains to

achieve efficiency improvements [20]. The work [21] delves into approximate computing techniques, exploring security issues and analyzing the impact of application level on neural network processing, as well as image, speech, and baseband signal processing. The work [22] specifically investigates the analysis of the impact of approximate computing on application quality through a three-step process involving error characterization, error propagation, and linking errors with the quality metric of the application. The work [23] explores the integration of approximation techniques with conventional computing tasks to enhance the efficient utilization of computational infrastructure. By characterizing a library of approximated operators, the work [24] proposes a Bayesian model for predicting error propagation. Their work demonstrates improved accuracy evaluations and computational efficiency, positioning it as a valuable tool for design space exploration in approximate computing.

In the context of next-generation communication systems, which prioritize flexible performance targets for enhanced energy efficiency [5,25], the use of approximate computing techniques becomes relevant, which is another approach for improving energy efficiency. These techniques allow a controlled trade-off in system performance, contributing to improved efficiency in communication systems. A comprehensive survey [26] on the potential of approximate computing techniques for existing and future B5G communication highlights SER as a crucial Key Performance Indicator (KPI) for channel-related problems while energy efficiency as a prime KPI for resource allocation. By employing approximate computing techniques within a fixed power budget for communication systems, it becomes possible to reduce overall system power consumption. This reduction in power consumption consequently frees up additional capacity for system scaling within the same power budget.

Recent advancements in approximate computing techniques for communication systems are summarized in Table 1. The decoder based on the Successive Cancellation (SC) algorithm enhances the Forward Error Correction (FEC) performance of polar codes; however, it limits the throughput of its hardware implementations. To tackle this challenge, configurable approximation units are introduced in optimized computation function blocks used in the SC algorithm to improve the throughput of the decoder in the work [27]. The work in [28] harnesses the error-resilient nature of the inherent Fast Fourier Transform (FFT) operation in the industrial wireless communication system to demonstrate the potential of approximate computing. The exact add/subtract operators in the butterfly structure of the FFT are replaced with approximate adders, and the impact of the modified FFT operations is analyzed at the system level. Additionally, the work emphasizes the challenges related to system reliability and suitable error metrics, stressing the need to establish a connection between the characteristics of approximate adders and system performance. The work in [29] explores the application of approximate computing in the expectation propagation algorithm used for the Sparse Code Multiple Access (SCMA) receiver. By employing approximation techniques, the complexity of the algorithm is reduced, which is characterized by the number of arithmetic operations. Approximations are incorporated into the expectation propagation algorithm at the variable and function node updates, as well as the log-likelihood ratio calculation, to decrease algorithmic complexity. Moreover, parameter optimizations are proposed to strike a balance between detection performance and algorithm complexity. In the work [30], exact computing units are substituted with approximate ones in Root Raised Cosine (RRC) Finite Impulse Response (FIR) filters used for pulse shaping at the BS and decoders/equalizers at the User Equipment (UE) in Single Input Single Output (SISO) and Multiple Input Multiple Output (MIMO) 6G downlink operations. The BER performance of the proposed approximate computing-empowered 6G SISO downlink is superior to its MIMO counterpart, where the induced approximations achieve substantial power savings. The BER performance degradation is more pronounced in the high SNR regime compared to the low/medium SNR regime. The work in [31] exploits gradient bounds to propose a novel encoding scheme for Quadrature Amplitude Modulation (QAM) mapping in the communication system required for a

federated learning model. The results highlight the significance of quantifying the effects of approximation on the overall application. In a fixed SNR scenario, the test accuracy of the model deteriorates as the QAM modulation order increases.

**Table 1.** Approximate computing techniques for communication systems.

| Work | Description | Approximation | Modulation | QoS |
|---|---|---|---|---|
| Zhou (2018) [27] | Throughput improvement by utilizing approximate computation blocks for decoding FEC polar codes. | SC decoder. | - | FER |
| Hao (2019) [28] | Reliability assessment on utilizing approximate adders for industrial wireless communication. | FFT. | QPSK | FER |
| Xiao (2019) [29] | Complexity reduction of expectation propagation algorithm utilized for SCMA. | Variable and functional node update, log likelihood ratio computation. | - | BER |
| Idrees (2021) [30] | Gains by utilizing approximate computing units in digital signal processing filters in 6G downlink operation. | FIR filter at pulse shaping/equalization/decoding. | BPSK, QPSK, 8-PSK | BER (link), Structural Similarity Index and Correlation Coefficient (Image transmission) |
| Ma (2023) [31] | Approximate communication scheme for federated learning application. | Gradients. | QPSK, 16-QAM, 256-QAM | Test accuracy |

While approximation techniques can enhance performance at the expense of reduced accuracy, their application tends to diminish the reliability of the system. In the work [32], strategies for testing approximate circuits are delineated, emphasizing the critical role of reliability in the application of approximation techniques to any system. To address the reliability challenge, it is necessary to precisely estimate the accuracy of the approximate system in correlation to the level of induced approximation. The work [33] delves into approximation techniques grounded in the determinism of system accuracy and the granularity control provided by these techniques. The objective is to bolster the reliability of approximate systems by accurately estimating system accuracy reflecting the level of approximation.

Multiplication operations are pivotal in communication systems, influencing overall processing capability and the efficiency of handling complex mathematical transformations. Precision in these operations is vital for maintaining system accuracy, directly impacting signal fidelity during reception. Moreover, the resource-intensive nature of multiplication operations consumes substantial computational resources. In the context of energy efficient communication system design, particularly in environments with resource constraints and battery-powered devices, optimizing the efficiency of multiplication operations becomes paramount. Therefore, the introduction of approximations in multiplication operations presents an opportunity to improve the overall energy efficiency of communication systems.

Arithmetic functional units performing operations like multiplication inherently function as nonlinear static systems and approximating them introduces irregularities in the system. The challenge lies in the insufficient capacity of current statistical metrics to fully capture these irregularities arising from approximation, creating a barrier to the

widespread adoption of approximate arithmetic units as concerns about system reliability emerge. In one such attempt, the work [34] seeks to assess the impact of a nonlinear approximate adder on the application using statistical metrics. Accurately measuring the irregularities introduced in the computing system due to approximation is essential for enhancing the reliability of approximated systems and making more informed choices regarding the selection of approximation techniques.

Noise models are prevalent to model the cause of irregularities in the system, which may be deterministic or non-deterministic [35]. The peak value evaluation error method in wireless receivers was studied in the work [36] for urban noise impulses using past experimental data for different frequency bands, modulation schemes, and bit rates. Non-Gaussian noise was statistically modeled in the work [37] for signal processing applications. The work [38] provides an overview of impulse noise and its models, highlighting their similarities and differences in communication systems and by comparing the performance of single-carrier and multi-carrier communication systems under impulse noise. A computationally intensive Gaussian mixture model is employed to model the impulsive noise for computing analytical expression of SER [39].

In the systematic analysis depicted in Figure 1, the preliminary entities are derived from components. These entities are utilized to formulate primary metrics, while secondary metrics are derived from the primary ones. All analyses are conducted using primary and secondary metrics. The proposed work introduces an AMN model to characterize irregularities resulting from the approximate multiplication operation in wireless signal detection as shown in Figure 2. The TM structure is used as an approximate multiplication technique, while the MMSE technique is utilized for signal detection in the proposed work. Utilizing the AMN, a closed-loop expression for the SER is derived for QPSK MMSE signal detection with TM. This expression serves to assess the resilience of various configurations of TM for MMSE detection.

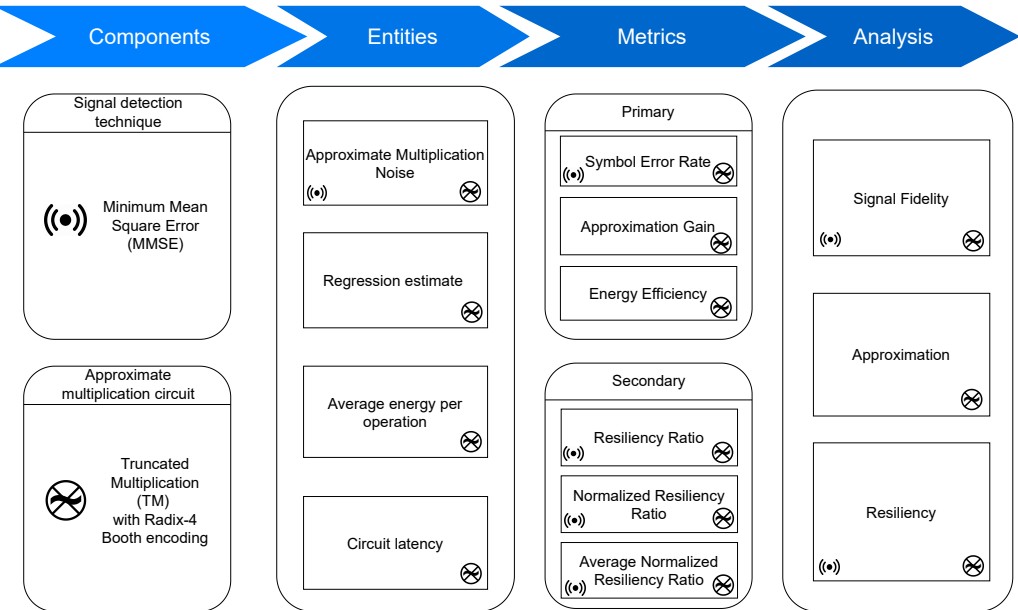

**Figure 1.** System analysis for energy-efficient wireless signal detection using approximate multiplication circuit. The relationship between entities, metrics, and analysis with components is explicitly demonstrated.

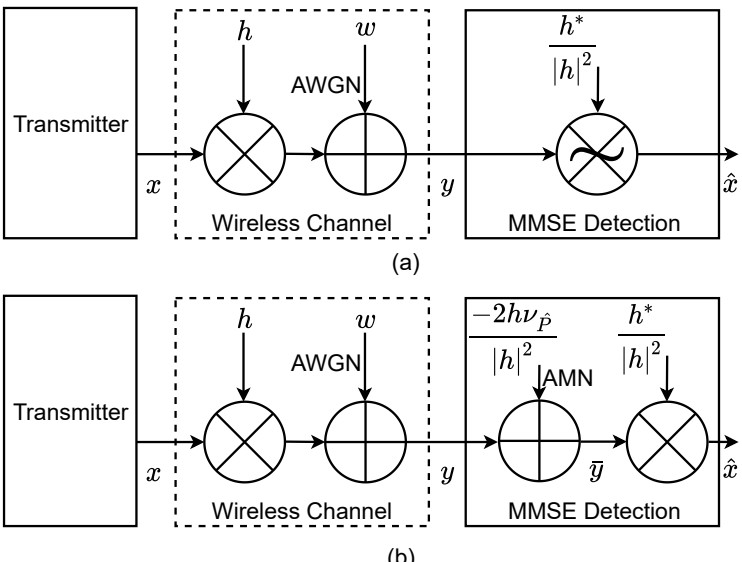

**Figure 2.** (**a**) MMSE detection using approximate multiplication. (**b**) Equivalent model for computing $\hat{x}$ using AMN and accurate multiplication.

## 3. Methodology

### 3.1. Preliminary

The notation $|.|$ signifies the absolute value, while $\mathbb{E}$ denotes the expectation operator. The symbols $\Re$ and $\Im$ represent the real and imaginary components, respectively. $\mathcal{P}\{\mathcal{G}\}$ denotes the probability of occurrence of any event $\mathcal{G}$. $[.]_2$ denotes integer values represented in two's complement form. The Probablity Distribution Function (PDF) of normal distribution with mean $\mu$ and variance $\sigma^2$ is represented as $\mathcal{N}(\mu, \sigma) = \frac{1}{\sigma\sqrt{2\pi}}\exp\left(-\frac{1}{2}\left(\frac{x-\mu}{\sigma}\right)^2\right)$, where $\exp(.)$ is the exponential function. Adding a constant $k$ to $\mathcal{N}(\mu, \sigma)$ results in a new distribution, $\mathcal{N}(k + \mu, \sigma)$. Multiplying a normal distribution by $k$ yields new normal distribution $\mathcal{N}(k\mu, k\sigma)$. For two normal distributions $\mathcal{N}(\mu_1, \sigma_1)$ and $\mathcal{N}(\mu_2, \sigma_2)$, the resultant normal distribution is given as $\mathcal{N}(\mu_1, \sigma_1) + \mathcal{N}(\mu_2, \sigma_2) = \mathcal{N}(\mu_1 + \mu_2, \sqrt{\sigma_1^2 + \sigma_2^2})$. The complementary error function, $\text{erfc}(.)$, evaluates to 0 as $\infty$ is approached and to 2 as $-\infty$ is approached. Using the symmetry property of $\text{erfc}(.)$, it can be inferred that $\text{erfc}(-x) = 2 - \text{erfc}(x)$. The integration of a normal distribution within the interval $[a, b]$ is expressed as the definite integral [40]:

$$\int_a^b \mathcal{N}(\mu, \sigma) = \frac{1}{2}\left(\text{erfc}\left(\frac{a - \mu}{\sigma\sqrt{2}}\right) - \text{erfc}\left(\frac{b - \mu}{\sigma\sqrt{2}}\right)\right) \tag{1}$$

$\text{Cov}(.)$ represents the covariance operation. The linear operations performed on a complex random variable apply to its real and imaginary components. The symbol $\asymp$ is used to denote the approximate multiplication of two operands.

### 3.2. Truncated Multiplication

Consider bit signals $a_i, b_i, \phi_i \in \{0, 1\}$ for $i = 0, 1, \ldots, N - 1$. In the context of signed multiplication, the $N$-bit multiplicand and multiplier operands are represented in two's complement form as $[A]_2 = -a_{N-1}2^{N-1} + \sum_{i=0}^{N-2} a_i 2^i$ and $[B]_2 = -b_{N-1}2^{N-1} + \sum_{i=0}^{N-2} b_i 2^i$, respectively. The multiplication product is denoted as $[P]_2 = -\phi_{2N-1}2^{2N-1} + \sum_{i=0}^{2N-2} \phi_i 2^i$. In the Radix-4 Booth algorithm for encoding the multiplier [41], the multiplication product $[P]_2$ is computed using the Algorithm 1.

---
**Algorithm 1** Signed multiplication using Radix-4 Booth algorithm.

---
1: $[A]_2$ and $[B]_2$ are $N$-bit multiplicand and multiplier.
2: **procedure** $([A]_2, [B]_2, N)$
3:      Initialize product $[P]_2$ as 0 with $2N$ bits.
4:      Initialize $b_{-1}$ as 0.
5:      **for** $i \leftarrow 0$ to $N/2 - 1$ **do**                            ▷ $N^2$ clock cycles
6:          **if** $b_{2i+1}b_{2i}b_{2i-1} = 001$ **or** $b_{2i+1}b_{2i}b_{2i-1} = 010$ **then**
7:             $[P]_2 \leftarrow [P]_2 + ([A]_2 \ll 2i)$              ▷ $2N$ clock cycles
8:          **else if** $b_{2i+1}b_{2i}b_{2i-1} = 101$ **or** $b_{2i+1}b_{2i}b_{2i-1} = 110$ **then**
9:             $[P]_2 \leftarrow [P]_2 - ([A]_2 \ll 2i)$              ▷ $2N$ clock cycles
10:         **else if** $b_{2i+1}b_{2i}b_{2i-1} = 011$ **then**
11:            $[P]_2 \leftarrow [P]_2 + ([A]_2 \ll 4i)$             ▷ $2N$ clock cycles
12:         **else if** $b_{2i+1}b_{2i}b_{2i-1} = 100$ **then**
13:            $[P]_2 \leftarrow [P]_2 - ([A]_2 \ll 4i)$             ▷ $2N$ clock cycles
14:         **end if**
15:      **end for**
16:      **return** $[P]_2$
17: **end procedure**

---

The multiplication operation is approximated for TM by truncating the $M$ least significant bits of every partial product generated by the multiplication operation, as illustrated in Figure 3. Thus, a TM configuration is decided by $M$. The TM product is obtained as $[\hat{P}]_2 = -\phi_{2N-1}2^{2N-1} + \sum_{i=M}^{2N-2} \phi_i 2^i$.

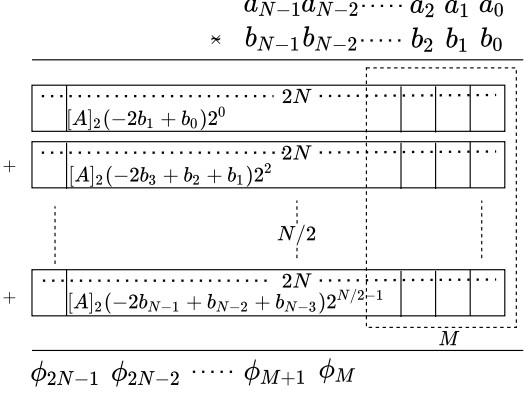

**Figure 3.** TM implemented using Radix-4 Booth algorithm for signed $N$ bit operands $[A]_2$ and $[B]_2$.

However, the operands used in baseband processing are rational numbers with a fractional part. Therefore, $N$-bit signed multiplication with fixed-point representation is considered for baseband processing, with $N/2$ bits allocated for the integer part and $N/2$ bits for the fractional part. Consequently, the operands for baseband processing are scaled by a factor of $2^{N/2}$ to convert them into two's complement form. The multiplication product is then converted back to fixed-point representation from two's complement form by using a scaling factor of $2^{-N}$. With operands $[A]_2 = A \cdot 2^{N/2}$ and $[B]_2 = B \cdot 2^{N/2}$, the accurate multiplication is represented as $P = ([A]_2[B]_2)2^{-N}$, while the TM product is represented as $\hat{P} = ([A]_2 \bowtie [B]_2)2^{-N}$.

The mean of the error between accurate multiplication and TM is given by $\mu_{\hat{P}} = \mathbb{E}\{[A]_2[B]_2\} - \mathbb{E}\{[A]_2 \bowtie [B]_2\}$. Considering the two's complement form of operands, the exact value of $\mu_{\hat{P}}$ is obtained through exhaustive simulation of all possible multiplication values for a particular $N$, as given in Table 2. For exhaustive simulation, all integer values from range $[-2^{N-1}, 2^{N-1} - 1)$ are considered for multiplicand and multiplier operand. However, when it is not feasible to perform exhaustive simulation for a particular $N$, $\mu_{\hat{P}}$ is point estimated with selective multiplication values. For point estimating $\mu_{\hat{P}}$, $K_N$ samples

from range $[-2^{N-1}, 2^{N-1} - 1)$ are considered for multiplicand and multiplier operand with an equal step size of $\frac{2^{N-1}-1}{K_N/2-1} = \frac{2^N-2}{K_N-2}$. The mean of the error for fixed-point representation is given by:

$$\nu_{\hat{P}} = \mathbb{E}\{P - \hat{P}\} = \mathbb{E}\{P\} - \mathbb{E}\{\hat{P}\} = \mu_{\hat{P}} 2^{-N} \tag{2}$$

**Table 2.** Error mean $\nu_{\hat{P}}$ values for $N = 8, 12, 16, 20$. $\nu_{\hat{P}}$ values for $N = 8, 12$ are calculated with exhaustive simulation of operands, while $\nu_{\hat{P}}$ values for $N = 16, 20$ are point estimated using $K_N = 4096$ and represented by †.

| | $\nu_{\hat{P}}$ | | | |
|---|---|---|---|---|
| $M$ | $N = 8$ | $N = 12$ | $N=16$ † | $N=20$ † |
| 1 | $9.77 \times 10^{-4}$ | $6.10 \times 10^{-5}$ | $3.81 \times 10^{-6}$ | $2.38 \times 10^{-7}$ |
| 2 | $3.91 \times 10^{-3}$ | $2.44 \times 10^{-4}$ | $1.57 \times 10^{-5}$ | $9.56 \times 10^{-7}$ |
| 3 | $1.37 \times 10^{-2}$ | $8.54 \times 10^{-4}$ | $5.64 \times 10^{-5}$ | $3.35 \times 10^{-6}$ |
| 4 | $3.71 \times 10^{-2}$ | $2.32 \times 10^{-3}$ | $1.55 \times 10^{-4}$ | $9.09 \times 10^{-6}$ |
| 5 | $9.96 \times 10^{-2}$ | $6.23 \times 10^{-3}$ | $4.13 \times 10^{-4}$ | $2.44 \times 10^{-5}$ |
| 6 | $2.40 \times 10^{-1}$ | $1.50 \times 10^{-2}$ | $9.88 \times 10^{-4}$ | $5.89 \times 10^{-5}$ |
| 7 | $5.84 \times 10^{-1}$ | $3.65 \times 10^{-2}$ | $2.38 \times 10^{-3}$ | $1.43 \times 10^{-4}$ |
| 8 | $1.33 \times 10^{0}$ | $8.34 \times 10^{-2}$ | $5.41 \times 10^{-3}$ | $3.27 \times 10^{-4}$ |
| 9 | - | $1.93 \times 10^{-1}$ | $1.24 \times 10^{-2}$ | $7.56 \times 10^{-4}$ |
| 10 | - | $4.27 \times 10^{-1}$ | $2.75 \times 10^{-2}$ | $1.67 \times 10^{-3}$ |
| 11 | - | $9.58 \times 10^{-1}$ | $6.15 \times 10^{-2}$ | $3.76 \times 10^{-3}$ |
| 12 | - | $2.08 \times 10^{0}$ | $1.33 \times 10^{-1}$ | $8.16 \times 10^{-3}$ |
| 13 | - | - | $2.93 \times 10^{-1}$ | $1.80 \times 10^{-2}$ |
| 14 | - | - | $6.27 \times 10^{-1}$ | $3.85 \times 10^{-2}$ |
| 15 | - | - | $1.36 \times 10^{0}$ | $8.35 \times 10^{-2}$ |
| 16 | - | - | $2.88 \times 10^{0}$ | $1.77 \times 10^{-1}$ |
| 17 | - | - | - | $3.81 \times 10^{-1}$ |
| 18 | - | - | - | $8.04 \times 10^{-1}$ |
| 19 | - | - | - | $1.71 \times 10^{0}$ |
| 20 | - | - | - | $3.59 \times 10^{0}$ |

### 3.2.1. Regression Estimate for $\hat{P}$

The $\hat{P}$ values are obtained through the operation of TM. To derive an analytical expression for the SER, there is a necessity for a mathematical model representing $\hat{P}$. The $\hat{P}$ values exhibit slight deviation from the $P$ values for a specific operand pair. This relationship between approximate and accurate multiplication values is depicted in Figure 4 for $N = 8$, where $P$ and $\hat{P}$ values display linear co-variation. This inference can be extended to other values of $N$ without loss of generality. Utilizing this sufficient statistical information about covariance, it is possible to estimate $\hat{P}$ from $P$ as the predictor variable in a linear regression model [42].

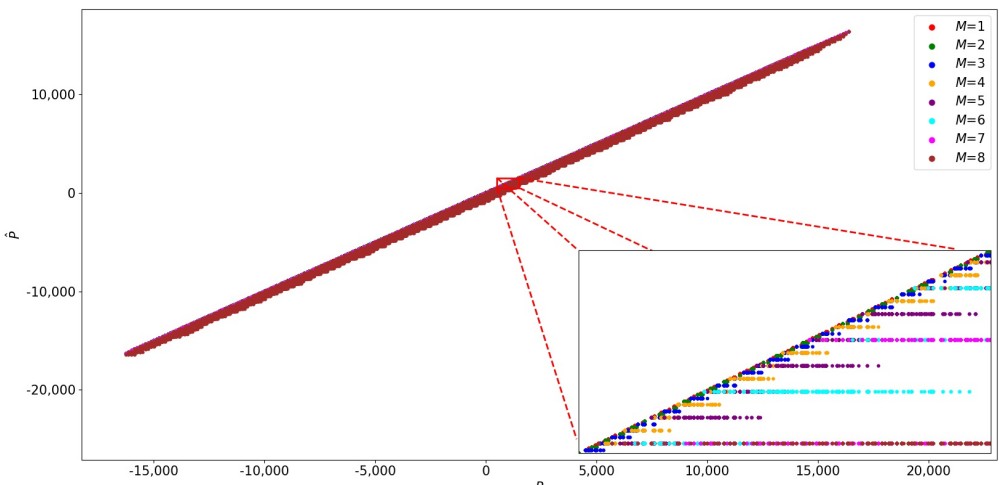

**Figure 4.** Variance of accurate multiplication values $P$ and TM values $\hat{P}$ for $N = 8$ for varying $M$. $P$ and $\hat{P}$ have linear covariance.

Consider a linear regression model $\hat{P} = \beta_0 + \beta_1 P + \epsilon$, where $\beta_0$ and $\beta_1$ are regression coefficients and $\epsilon \sim \mathcal{N}(0, \sigma^2)$ is the error term. The estimated regression model is given as $\hat{P} \approx \beta_0 + \beta_1 P$ and the values of estimated regression coefficients $\hat{\beta_0}$ and $\hat{\beta_1}$ are calculated as follows:

$$\hat{\beta_1} = \frac{\text{Cov}(\hat{P}, P)}{\text{Cov}(P, P)} \tag{3}$$

and

$$\hat{\beta_0} = \mathbb{E}\{\hat{P}\} - \hat{\beta_1}\mathbb{E}\{P\} \tag{4}$$

Since $\text{Cov}(\hat{P}, P) \approx \text{Cov}(P, P)$ as inferred from Figure 4, it implies that $\hat{\beta_1} \approx 1$; hence, $\hat{\beta_0}$ can be evaluated from Equation (4) and Equation (2) as follows:

$$\hat{\beta_0} = \mathbb{E}\{\hat{P}\} - \mathbb{E}\{P\} = -\nu_{\hat{P}} \tag{5}$$

The estimated regression model can be calculated as follows:

$$\hat{P} \approx P - \nu_{\hat{P}} \tag{6}$$

The linear regression model for $\hat{P}$ in Equation (6) provides a crude estimate of the multiplication product values of TM, which will be used for SER computation.

### 3.2.2. Energy Efficiency

The energy efficiency of TM for a particular $N$ and $M$ is computed with respect to the accurate multiplication. For the analysis of energy efficiency, an approximate expression is computed henceforth. For any multiplication operation performed using a digital logic processor, the total power consumed can be expressed as $E \cdot \#$ multiplication/sec, where $E$ is the average energy consumed per multiplication operation. The latency $T$ of the multiplication operation is computed as the product of total clock cycles and the processor clock frequency. The number of multiplication operations per second is thus $\frac{1}{T}$. Low $E$ signifies a more energy-efficient multiplication operation.

For accurate multiplication operation using the Radix-4 Booth algorithm, the total clock cycles consumed by the multiplication operation stem from the steps outlined in Algorithm 1. Primarily, every partial product computation consumes about $2N$ clock cycles

for a typical ripple carry addition operation. As there are $N/2$ partial products to be computed, the total clock cycles consumed are about $N/2(2N) = N^2$. In the case of TM as shown in Figure 3, the operand bit-width for partial product addition is $2N - M$. Hence, the total clock cycles for TM are:

$$\frac{N}{2}(2N - M) = N^2 - \frac{NM}{2} \tag{7}$$

For a processor operating at a clock frequency of $F$ Hz, the latency of accurate multiplication and TM is computed as:

$$T_P = N^2 F, \ T_{\hat{P}} = \left( N^2 - \frac{NM}{2} \right) F \tag{8}$$

For a power budget of $Q$ Joule intended for multiplication operation on the processor, consider both accurate multiplication and TM for a specific task. Then, the relation between energy per operation for accurate multiplication denoted as $E_P$ and $E_{\hat{P}}$ for TM can be derived as follows:

$$Q = \frac{E_P}{T_P}, \quad Q = \frac{E_{\hat{P}}}{T_{\hat{P}}} \implies \frac{E_P}{T_P} = \frac{E_{\hat{P}}}{T_{\hat{P}}} \tag{9}$$

To quantify the reduction in average energy consumed per multiplication operation, the fractional gain due to approximation caused by TM can be computed as approximation gain as follows:

$$\text{Approximation Gain} = \frac{E_P}{E_{\hat{P}}} = \frac{T_P}{T_{\hat{P}}} = \frac{N^2 F}{\left( N^2 - \frac{NM}{2} \right) F} = \frac{2N}{2N - M} \tag{10}$$

The relative change in $E_{\hat{P}}$ with respect to $E_P$ is calculated as:

$$\frac{E_P - E_{\hat{P}}}{E_P} = 1 - \frac{E_{\hat{P}}}{E_P} \tag{11}$$

Expressing Equation (11) in terms of $N$ and $M$ as:

$$\text{Energy Efficiency} = \frac{E_P - E_{\hat{P}}}{E_P} = \frac{M}{2N} \tag{12}$$

The expressions for approximation gain and energy efficiency enable us to compare the gains achieved due to approximation for multiplication operation with accurate multiplication and TM.

### 3.3. SER Expression

For the QPSK constellation of symbols with each symbol represented by 2 bits, the alphabets for symbols are represented by set $\mathcal{X} = \{-1 - j1, -1 + j1, 1 - j1, 1 + j1\}$. Let $x$ be the signal representing the transmit symbol such that $x \in \mathcal{X}$ considering Additive White Gaussian Noise (AWGN) channel and constant fading channel $h$. $E_s$ represents the energy per symbol. AWGN for the receiver system is represented by $w$ and the random variable $W$ is used to represent the values of $w$ such that $W \sim \mathcal{CN}(0, \sigma_n^2)$, where $\sigma_n^2 = N_0$ and $N_0$ being the Noise Spectral Density (NSD). $E_s/N_0$ is used to denote the SNR. The noise variance is evenly distributed between real and imaginary components of the symbol such that the variance per component is $\frac{\sigma_n^2}{2}$. The scaling factor for energy normalization for $x$ is given as $S = \sqrt{\frac{E_s}{2}}$.

The received signal is represented as:

$$y = Shx + w \tag{13}$$

Consider AMN as $\delta = \delta_\Re + j\delta_\Im$. The received signal with AMN:

$$\overline{y} = Shx + w + \delta = y + \delta \tag{14}$$

MMSE detection [43] is a signal detection technique derived using optimization of the mean square error between transmit and receive symbols. MMSE detection achieves near-optimal performance while maintaining low computational complexity, rendering it highly suitable for implementation across a wide array of computing platforms, particularly those with limited computing resources. Additionally, MMSE detection exhibits high robustness across various SNR conditions, particularly in scenarios characterized by low SNR levels. This robustness enhances error resilience within the system, which enables expansion of the scope for approximation with high reliability.

The transmit symbol is estimated using MMSE at the receiver represented by $\hat{x}$ such that:

$$\hat{x} = \frac{1}{S} \frac{\mathbb{E}\{|x|^2\}}{\mathbb{E}\{|x|^2\}|h|^2 + N_0} h^* y = \frac{h^*}{S|h|^2} y \tag{15}$$

as $\mathbb{E}\{|x|^2\}|h|^2 \gg N_0$.

### 3.3.1. AMN Model

The MMSE computes $\hat{x}$ for transmit symbol $x$; however, by employing TM at the detection stage, an approximated $\hat{x}$ is obtained. To compute expression for $\delta$, the signal $\hat{x}$ approximated by using TM at the receiver as shown in Figure 2a is equated to equivalent model of $\hat{x}$ approximated by employing AMN and accurate multiplication as shown in Figure 2b. The approximate value of $\hat{x}$ using TM at the receiver is computed as follows:

$$
\begin{aligned}
\hat{x} &\approx \frac{h^*}{S|h|^2} \approx y = \frac{1}{S|h|^2}(h_\Re \approx y_\Re + h_\Im \approx y_\Im) + \frac{j}{S|h|^2}(h_\Re \approx y_\Im - h_\Im \approx y_\Re) \\
&\approx \left( \frac{h_\Re}{S|h|^2} y_\Re + \frac{h_\Im}{S|h|^2} y_\Im - \frac{2v_{\hat{P}}}{S|h|^2} \right) + j \left( \frac{h_\Re}{S|h|^2} y_\Im - \frac{h_\Im}{S|h|^2} y_\Re \right)
\end{aligned}
\tag{16}
$$

The approximate value of $\hat{x}$ using accurate multiplication by employing AMN is computed as follows:

$$
\begin{aligned}
\hat{x} &\approx \frac{h^*}{S|h|^2}\overline{y} = \left( \frac{h_\Re}{S|h|^2}\overline{y}_\Re + \frac{h_\Im}{S|h|^2}\overline{y}_\Im \right) + j\left( \frac{h_\Re}{S|h|^2}\overline{y}_\Im - \frac{h_\Im}{S|h|^2}\overline{y}_\Re \right) \\
&\approx \left( \frac{h_\Re}{S|h|^2}y_\Re + \frac{h_\Im}{S|h|^2}y_\Im + \frac{h_\Re\delta_\Re + h_\Im\delta_\Im}{S|h|^2} \right) + \\
&\quad j\left( \frac{h_\Re}{S|h|^2}y_\Im - \frac{h_\Im}{S|h|^2}y_\Re + \frac{h_\Re\delta_\Im - h_\Im\delta_\Re}{S|h|^2} \right)
\end{aligned}
\tag{17}
$$

Equating Equations (16) and (17):

$$h_\Re\delta_\Re + h_\Im\delta_\Im = -2v_{\hat{P}} \tag{18}$$

and

$$h_\Re\delta_\Im - h_\Im\delta_\Re = 0 \tag{19}$$

From Equations (18) and (19), the values of $\delta_\Re$, $\delta_\Im$ and eventually $\delta$ can be evaluated as:

$$\delta = \frac{-2v_{\hat{P}}h_\Re}{|h|^2} - j\frac{2v_{\hat{P}}h_\Im}{|h|^2} = \frac{-2hv_{\hat{P}}}{|h|^2} \tag{20}$$

AMN facilitates use of signal $\overline{y}$ in $\hat{x}$ detection.

### 3.3.2. SER Evaluation Using AMN

With signal $x$, consider the approximate detection of $\hat{x}$ using MMSE detection by using signal $\overline{y}$. For AWGN channel, the random variable for $\overline{y}$ is modeled as $\overline{Y} = Shx + W + \delta$.

Separating $\overline{Y}$ into real and imaginary components $\overline{Y}_\Re \sim \mathcal{N}(Sh_\Re x_\Re - Sh_\Im x_\Im + \delta_\Re, \sigma_n/\sqrt{2})$ and $\overline{Y}_\Im \sim \mathcal{N}(Sh_\Re x_\Im + Sh_\Im x_\Re + \delta_\Im, \sigma_n/\sqrt{2})$. Hence,

$$\hat{x} \approx \frac{h^*}{S|h|^2}\overline{y} \implies \hat{X} \approx \frac{h^*}{S|h|^2}\overline{Y} \tag{21}$$

$$\hat{X} \approx \left(\frac{h_\Re}{S|h|^2}\overline{Y}_\Re + \frac{h_\Im}{S|h|^2}\overline{Y}_\Im\right) + j\left(\frac{h_\Re}{S|h|^2}\overline{Y}_\Im - \frac{h_\Im}{S|h|^2}\overline{Y}_\Re\right) \tag{22}$$

The linear combination of two random variables also results in a random variable [44]. Using the PDF for $\overline{Y}_\Re$ and $\overline{Y}_\Im$ in Equation (22) and $\delta$, the PDF expressions for real and imaginary parts for $\hat{X}$ are evaluated as follows.

$$
\begin{aligned}
f_{\hat{X}_\Re} &\approx \frac{1}{S|h|^2}\mathcal{N}\left(h_\Re(Sh_\Re x_\Re - Sh_\Im x_\Im + \delta_\Re) + h_\Im(Sh_\Re x_\Im + Sh_\Im x_\Re + \delta_\Im), \frac{\sigma_n}{\sqrt{2}}\sqrt{h_\Re^2 + h_\Im^2}\right) \\
&\approx \frac{1}{S|h|^2}\mathcal{N}\left(x_\Re S(h_\Re^2 + h_\Im^2) - 2\,v_{\hat{p}}, \frac{\sigma_n}{\sqrt{2}}|h|\right) \\
&\approx \mathcal{N}\left(x_\Re - \frac{2v_{\hat{p}}}{S|h|^2}, \frac{\sigma_n}{\sqrt{2}S|h|}\right)
\end{aligned} \tag{23}
$$

$$
\begin{aligned}
f_{\hat{X}_\Im} &\approx \frac{1}{S|h|^2}\mathcal{N}\left(h_\Re(Sh_\Re x_\Im + Sh_\Im x_\Re + \delta_\Im) - h_\Im(Sh_\Re x_\Re - Sh_\Im x_\Im + \delta_\Re), \frac{\sigma_n}{\sqrt{2}}\sqrt{h_\Re^2 + h_\Im^2}\right) \\
&\approx \frac{1}{S|h|^2}\mathcal{N}\left(x_\Im S(h_\Re^2 + h_\Im^2), \frac{\sigma_n}{\sqrt{2}}\sqrt{h_\Re^2 + h_\Im^2}\right) \\
&\approx \mathcal{N}\left(x_\Im, \frac{\sigma_n}{\sqrt{2}S|h|}\right)
\end{aligned} \tag{24}
$$

To compute the area under the inference regions for QPSK as shown in Figure 5, consider integrals $I1$, $I2$, $I3$, and $I4$. Using $f_{\hat{X}_\Re}$ and $f_{\hat{X}_\Im}$ when $x_\Re, x_\Im = \pm 1$, these integrals are computed as follows:

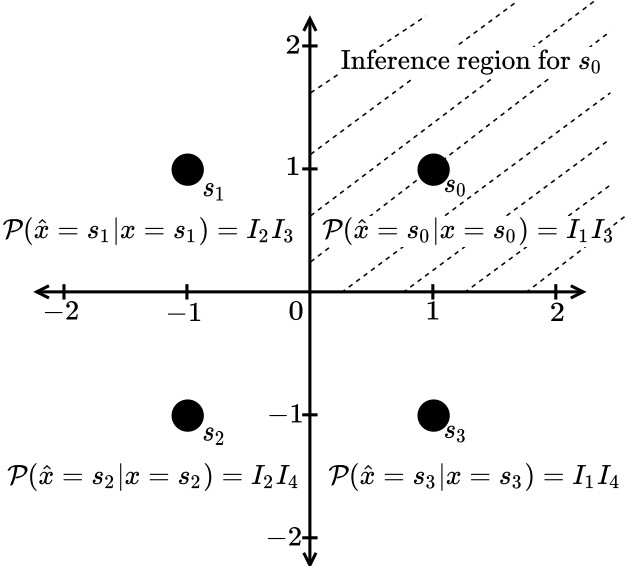

**Figure 5.** Inference regions for the QPSK constellation.

$$I1 = \int_0^\infty f_{\hat{X}_\Re}\Big|_{x_\Re = 1} = \int_0^\infty \mathcal{N}\left(1 - \frac{2v_{\hat{P}}}{S|h|^2}, \frac{\sigma_n}{\sqrt{2}S|h|}\right)$$

$$= \frac{1}{2}\text{erfc}\left(\frac{-S|h|^2 + 2v_{\hat{P}}}{\sigma_n|h|}\right) \tag{25}$$

$$I2 = \int_{-\infty}^0 f_{\hat{X}_\Re}\Big|_{x_\Re = -1} = \int_{-\infty}^0 \mathcal{N}\left(-1 - \frac{2v_{\hat{P}}}{S|h|^2}, \frac{\sigma_n}{\sqrt{2}S|h|}\right)$$

$$= \frac{1}{2}\left(2 - \text{erfc}\left(\frac{S|h|^2 + 2v_{\hat{P}}}{\sigma_n|h|}\right)\right) = \frac{1}{2}\text{erfc}\left(\frac{-S|h|^2 - 2v_{\hat{P}}}{\sigma_n|h|}\right) \tag{26}$$

$$I3 = \int_0^\infty f_{\hat{X}_\Im}\Big|_{x_\Im = 1} = \int_0^\infty \mathcal{N}\left(1, \frac{\sigma_n}{\sqrt{2}S|h|}\right) = \frac{1}{2}\text{erfc}\left(\frac{-S|h|}{\sigma_n}\right) \tag{27}$$

$$I4 = \int_{-\infty}^0 f_{\hat{X}_\Im}\Big|_{x_\Im = -1} = \int_{-\infty}^0 \mathcal{N}\left(-1, \frac{\sigma_n}{\sqrt{2}S|h|}\right)$$

$$= \frac{1}{2}\left(2 - \text{erfc}\left(\frac{S|h|}{\sigma_n}\right)\right) = \frac{1}{2}\text{erfc}\left(\frac{-S|h|}{\sigma_n}\right) \tag{28}$$

These integrals are used in evaluating the probability of correct symbol detection by the receiver for every symbol in $\mathcal{X}$. The SER for QPSK is computed by evaluating the union probability that each of the symbols in $\mathcal{X}$ is transmitted and received correctly at the receiver [45]. The SER expression is computed as follows:

$$SER = 1 - \frac{1}{4}\sum_{i=1}^4 \mathcal{P}(\hat{x} = s_i | x = s_i)$$

$$= 1 - \frac{1}{4}(I_1 I_3 + I_2 I_3 + I_2 I_4 + I_1 I_4) = 1 - \frac{1}{4}(I3 + I4)(I1 + I2)$$

$$= 1 - \frac{1}{8}\text{erfc}\left(\frac{-S|h|}{\sigma_n}\right)\left(\text{erfc}\left(\frac{-S|h|^2 + 2v_{\hat{P}}}{\sigma_n|h|}\right) + \text{erfc}\left(\frac{-S|h|^2 - 2v_{\hat{P}}}{\sigma_n|h|}\right)\right) \tag{29}$$

Substituting the values of $S$ and $\sigma_n$,

$$= 1 - \frac{1}{8}\text{erfc}\left(-|h|\sqrt{\frac{E_s}{2N_0}}\right)\left(\text{erfc}\left(\frac{-|h|^2\sqrt{E_s} + 2\sqrt{2}v_{\hat{P}}}{|h|\sqrt{2N_0}}\right) + \right.$$

$$\left. \text{erfc}\left(\frac{-|h|^2\sqrt{E_s} - 2\sqrt{2}v_{\hat{P}}}{|h|\sqrt{2N_0}}\right)\right) \tag{30}$$

SER is computed by simulation in Figure 2a using 5000 symbols in Python. The analytical expression of Equation (30) tracks SER obtained using simulation for $N = 8, 16$ as shown in Figure 6. It can be observed that for both $N = 8$ and $N = 16$, the SER starts increasing with an increase in SNR after $M = N/2$.

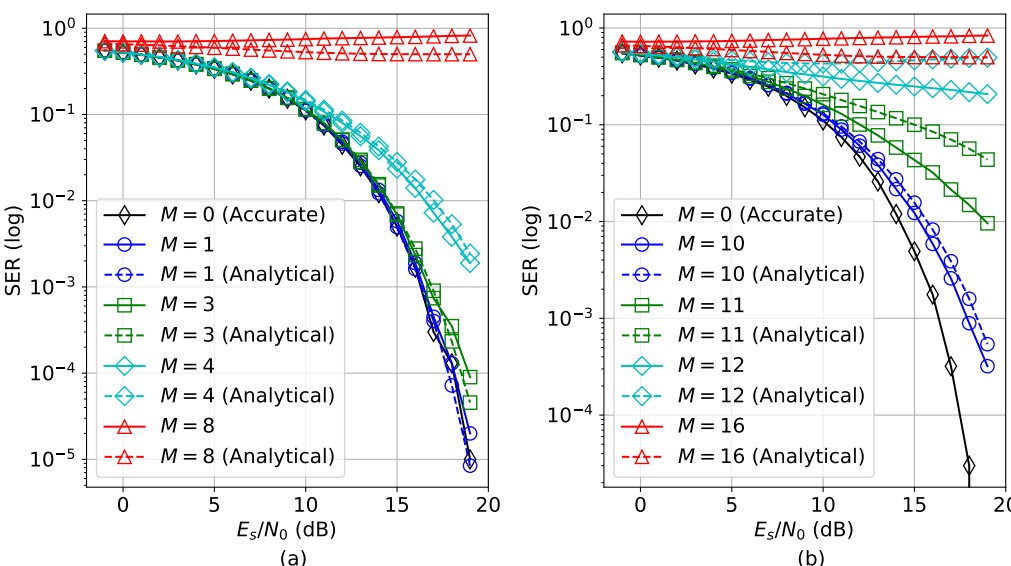

**Figure 6.** Comparison of SER for $N = 8, 16$ with channel gain $|h|^2 = 0.25$ for varying $M$ calculated analytically using Equation (30) and by simulation for (**a**) $N = 8$; (**b**) $N = 16$.

## 4. Analysis

### 4.1. Signal Fidelity Analysis

The SER of the QPSK MMSE signal detection using TM is evaluated in Figure 7 under varying channel gain conditions. The SER for a specific $N$, a specific channel gain and a specific SNR is bounded within an interval. The SER exhibits resilience for a range of $M$, forming the lower bound and begins to degrade thereafter with an increase in $M$. Furthermore, after a certain value of $M$, it becomes constant with a further increase in $M$, establishing the upper bound of SER.

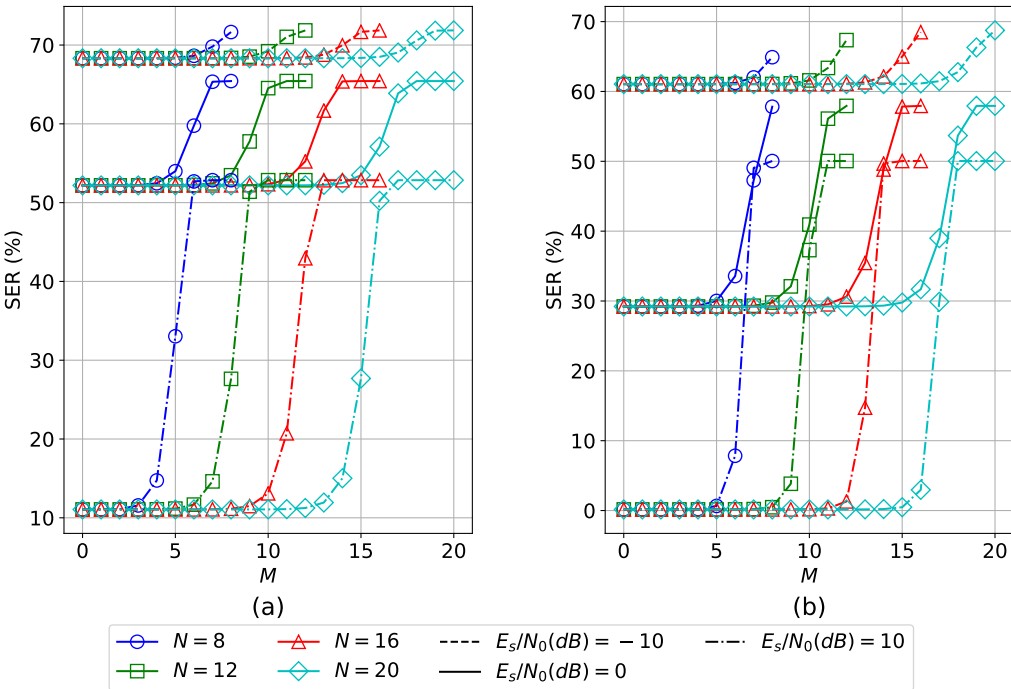

**Figure 7.** Signal fidelity analysis for QPSK MMSE signal detection using TM for varying $M$ for $N = 8, 16, 12$ and $20$ for channel gain (**a**) $|h|^2 = 0.25$; (**b**) $|h|^2 = 1$.

For a given channel gain, the bound interval increases with an increase in SNR. Although the lower bound of SER decreases with an increase in channel gain, the bound interval increases significantly. This implies that approximation causes more degradation of SER for high channel gain. However, the resiliency is sustained for higher values of $M$ when channel gain is increased for the same SNR. Additionally, resiliency is maintained for lower values of $M$, when SNR is increased for the same channel gain.

### 4.2. Approximation Analysis

TM configurations for $N = 8, 12, 16$ and 20 are analyzed for approximation gain and energy efficiency in Figure 8. Approximation gain is assessed using Equation (10) and energy efficiency is evaluated using Equation (12). Both approximation gain and energy efficiency increase with an augmentation in $M$ for all values of $N$. However, the rate of increase in approximation gain and energy efficiency decreases as $N$ increases. This observation implies that the approximation gain and energy efficiency are lower for higher values of $N$ for a specific value of $M$.

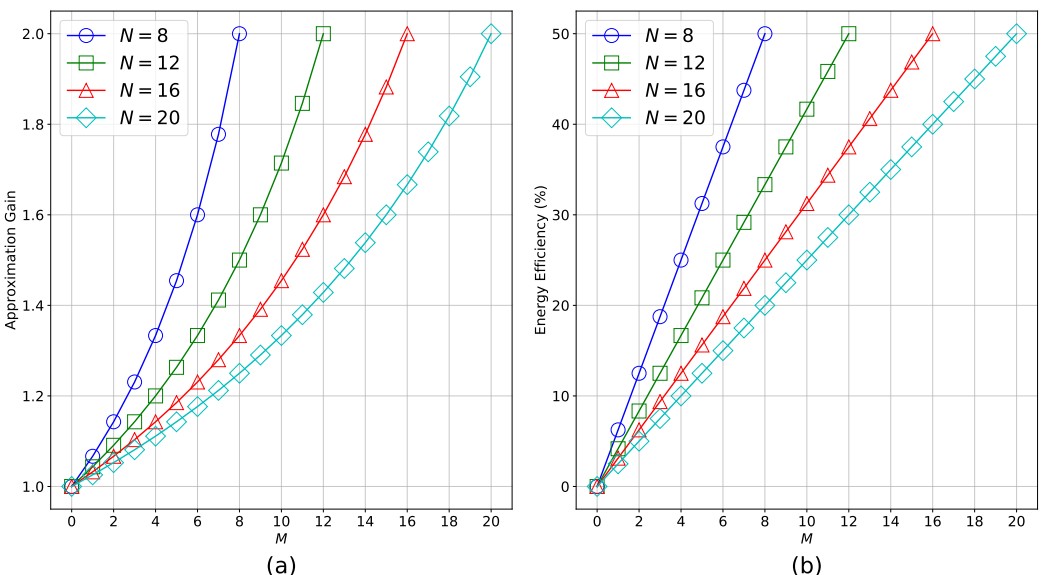

**Figure 8.** Approximation analysis with varying $M$ truncation bits: (**a**) Approximation Gain; (**b**) Energy efficiency.

### 4.3. Resiliency Analysis

For achieving high energy efficiency of the TM configuration in QPSK MMSE signal detection, it is crucial to choose a configuration that exhibits a high approximation gain for high energy efficiency. However, high approximation gain necessitates a corresponding increase in the value of $M$, resulting in a rise in the SER. Balancing a high approximation gain and minimizing the SER is key to achieving optimal performance when utilizing a TM for QPSK MMSE signal detection. The resiliency of TM configuration can be considered high if it provides high approximation gain without degrading much of the SER.

The Resiliency Ratio (RR) serves as a metric to quantify the resiliency of TM operation for QPSK MMSE detection for a specific combination of $M$, SNR and channel gain computed as follows:

$$\text{RR}\big|_{\text{SNR},|h|^2} = \frac{\text{Approximation Gain}}{SER\big|_{\text{SNR},|h|^2}} \tag{31}$$

The Normalized Resiliency Ratio (NRR), defined as the ratio of RR to the maximum RR for a given $h$ and SNR across all values of $M$ scales the RR in range of $[0, 1]$ and is computed as follows:

$$NRR\big|_{\text{SNR},|h|^2} = \frac{RR\big|_{\text{SNR},|h|^2}}{\max\limits_{M \leq N}\left(RR\big|_{\text{SNR},|h|^2}\right)} \tag{32}$$

While the NRR is a useful metric to analyze resiliency, it depends on SNR and channel gain. Hence, Average Normalized Resiliency Ratio (ANRR) provides insights into overall resiliency of the TM and is computed by averaging NRR values for $K_{\text{SNR}}$ samples of SNR in range $[-10, 10)$ and $K_{|h|}$ samples of channel gain in range of $[0, 1)$ as follows:

$$ANRR = \frac{1}{K_{\text{SNR}}K_{|h|}} \sum_{|h|=0}^{1} \sum_{\text{SNR}=-10}^{10} \left(NRR\big|_{\text{SNR},|h|^2}\right) \tag{33}$$

The graph of NRR and ANRR exhibits a distinct resiliency dip, commencing at the resiliency crest and concluding at the resiliency trough. An examination of Figure 9 reveals that for lower values of $M$, NRR tends to be lower for lower SNR regime, as depicted in Figure 9a,d, in contrast to its behavior in high SNR regime seen in Figure 9c,f. Conversely, at higher values of $M$, NRR significantly rises at low SNR compared to its performance at high SNR. The resilience of approximations with low $M$ is more pronounced in the high SNR regime, while those with high $M$ are more resilient in the low SNR regime. The extent of the resiliency dips expands with an increase in both SNR and channel gain. Also, the value of $M$ required to reach the resiliency crest rises with SNR.

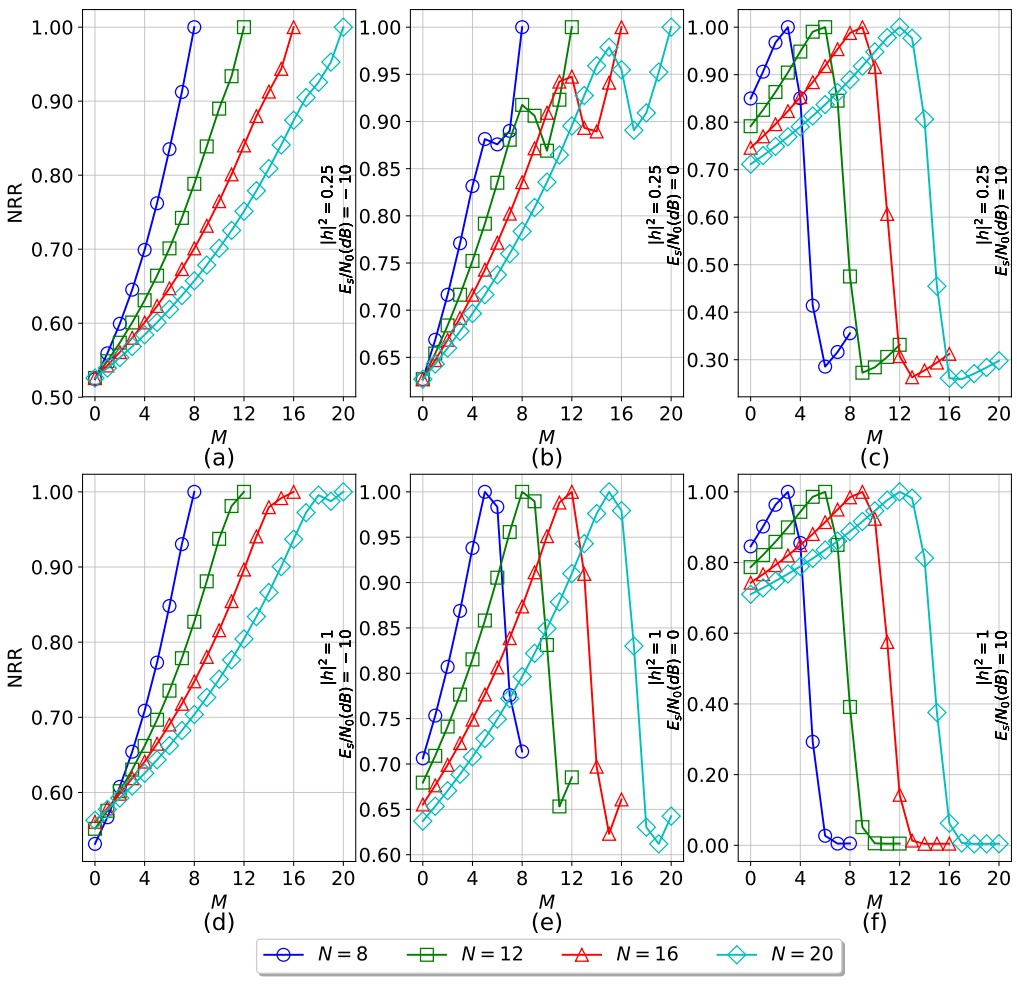

**Figure 9.** NRR analysis for QPSK MMSE signal detection using TM for varying $M$. NRR evaluated for (**a**) $|h|^2 = 0.25$, $E_s/N_0(dB) = -10$; (**b**) $|h|^2 = 0.25$, $E_s/N_0(dB) = 0$; (**c**) $|h|^2 = 0.25$, $E_s/N_0(dB) = 10$; (**d**) $|h|^2 = 1$, $E_s/N_0(dB) = -10$; (**e**) $|h|^2 = 1$, $E_s/N_0(dB) = 0$; (**f**) $|h|^2 = 1$, $E_s/N_0(dB) = 10$.

In the lower SNR regime, NRR experiences low resiliency dips, while dips begin to form with an increase in channel gain. NRR approaches 0 after the resiliency trough for high $M$ in high SNR regime and high channel gain. The resiliency trough decreases with an increase in SNR for all values of $N$. After a resiliency dip, surpassing the NRR beyond the resiliency crest towards 1 becomes increasingly difficult with an increase in SNR and channel gain. Until the resiliency crest is achieved, the rate of increase of NRR amplifies with an increase in both SNR and channel gain. Approximation with higher $M$ is advantageous in low SNR regime, while that with lower $M$ is advantageous in high SNR regime.

From the ANRR analysis presented in Figure 10 using $K_{\mathrm{SNR}} = 20$ and $K_{|h|} = 100$, it is evident that the rate of increase in ANRR diminishes with an increase in $N$. Resiliency dips for ANRR intensify with an increase in $N$ and the resiliency trough decreases with an increase in $N$.

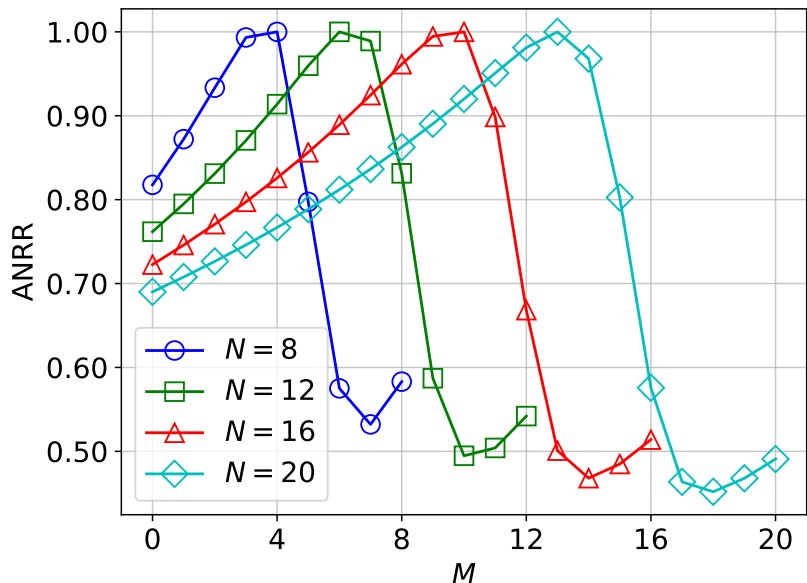

**Figure 10.** ANRR analysis for QPSK MMSE signal detection using TM for varying $M$ with $K_{\mathrm{SNR}} = 20$ and $K_{|h|} = 100$.

## 5. Conclusions

The proposed work systematically investigates the impact of employing approximate multiplication, represented by TM, on wireless signal detection, represented by QPSK MMSE detection. It aims to address the challenge of linking approximation level with QoS of the system through analyses of signal fidelity, resilience, and approximation. The study evaluates the AMN model to understand the effects of irregularities induced in the system by approximation and derives an analytical expression for the SER using the AMN model for signal fidelity analysis. A summary of all the simulations conducted for the proposed work is depicted in Table 3.

Signal fidelity analysis forecasts system output at different approximation levels, thereby enhancing the stability of the system. Energy efficiency increases with higher levels of approximation in TM. However, an increase in TM approximation leads to degradation of SER, and system reliability begins to decline after error resilience has been exhausted beyond a certain level of approximation. Resilience metrics capture this phenomenon and provide insights into reliable approximate configurations. Higher levels of approximation exhibit more resilience in low SNR scenarios, while lower levels of approximation are more resilient in high SNR scenarios. In the context of the proposed work, future avenues for exploration include extending the AMN model to accommodate high modulation schemes, diverse channel models, a variety of approximate multiplication schemes, and alternative receiver techniques.

**Table 3.** Simulation Summary.

| Analysis | Metric | Parameter | Description |
|---|---|---|---|
| **Entities** | | | |
| Computation | $\nu_{\hat{P}}$ | $N = 8, 12$; $M = 1 \ldots N$; $K_N = 2^N$; Operand range $[-2^{N-1}, 2^{N-1} - 1)$ | As given in Table 2, $\nu_{\hat{P}}$ values are computed by exhaustive simulation. |
| Computation | $\nu_{\hat{P}}$ | $N = 16, 20$; $M = 1 \ldots N$; $K_N = 4096$; Operand range: $[-2^{N-1}, 2^{N-1} - 1)$; step size $= \frac{2^N - 2}{K_N - 2}$ | As given in Table 2, $\nu_{\hat{P}}$ values are computed by point estimation. |
| Covariance | $P, \hat{P}$ | $N = 8$; $M = 1 \ldots N$; $K_N = 2^N$; Operand range $[-2^{N-1}, 2^{N-1} - 1)$ | As depicted in Figure 4, $P$ and $\hat{P}$ have linear covariance. |
| Verification | $SER$ | $N = 8, 16$; $M = 1 \ldots N$; $|h|^2 = 0.25$, 5000 symbols per $E_s/N_0(dB)$ | As depicted in Figure 6, the analytical expression for SER tracks the SER computed by simulation. |
| **System** | | | |
| Signal Fidelity | $SER$ | $N = 8, 12, 16, 20$; $M = 1 \ldots N$; $|h|^2 = 0.25, 1$; $E_s/N_0(dB) = -10, 0, 10$ | As depicted in Figure 7, the bound interval increases with both the SNR and channel gain, which indicates a greater degradation in SER. |
| Approximation | Approximation Gain | $N = 8, 12, 16, 20$; $M = 1 \ldots N$ | As shown in Figure 8a, approximation gain increases with $M$ for all $N$, while the rate of increase decreases with $N$. |
| Approximation | Energy Efficiency | $N = 8, 12, 16, 20$; $M = 1 \ldots N$ | As shown in Figure 8b, energy efficiency increases with $M$ for all $N$, while the rate of increase decreases with $N$. |
| Resiliency | NRR | $N = 8, 12, 16, 20$; $M = 1 \ldots N$; $|h|^2 = 0.25, 1$; $E_s/N_0(dB) = -10, 0, 10$ | As shown in Figure 9, high NRR is achieved by low $M$ in high SNR regime and high $M$ in low SNR regime. |
| Resiliency | ANRR | $N = 8, 12, 16, 20$; $M = 1 \ldots N$; $K_{\text{SNR}} = 20$; $K_{|h|} = 100$ | As shown in Figure 10, rate of increase in ANRR decreases with $N$ and resiliency dips increase with $N$. |

**Author Contributions:** Conceptualization, A.K.; methodology, A.K.; software, A.K.; validation, A.K.; formal analysis, A.K.; investigation, A.K.; resources, M.A.O. and D.M.; data curation, A.K.; writing—original draft preparation, A.K.; writing—review and editing, M.A.O. and D.M.; visualization, A.K.; supervision, M.A.O. and D.M.; project administration, M.A.O.; funding acquisition, D.M. All authors have read and agreed to the published version of the manuscript.

**Funding:** The research was funded in part by Natural Sciences and Engineering Research Council of Canada (NSERC): CRSNG-RDCPJ 514758-17, in part by Prompt, in part by Canadian Foundation for Innovation (CFI), in part by CMC Microsystems, in part by Opal-RT Technologies Inc. and in part by Hydro-Québec.

**Data Availability Statement:** Data are contained within the article.

**Conflicts of Interest:** The authors declare no conflicts of interest. The funders had no role in the design of the study; in the collection, analyses, or interpretation of data; in the writing of the manuscript; or in the decision to publish the results.

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
