# Peer review of "Energy Efficient Wireless Signal Detection: A Revisit through the Lens of Approximate Computing"

_electronics, doi:10.3390/electronics13071274_

Round 1
Reviewer 1 Report
Comments and Suggestions for Authors
Please answer the following questions and provide some details to enhance the quality of the manuscript:
1) The appearance of the constant noise model feels abrupt. What noise model is typically used in similar scenarios? Regarding the constant noise model mentioned here, what specific characteristic of the noise is considered constant? Can it accurately describe the noise characteristics of the system processing?
2) The author should put more effort into proofreading the formatting of chapter numbering, for example, the section numbering in the Outline section is labeled as section 3.1, while there is only one section in Chapter 3.
Another issue is that sections 1 through 3 should all belong to the Introduction section. The author's Section numbering in the text is confusing.
3) The author states in the abstract, "The analytical expression for Symbol Error Rate (SER) is evaluated using a constant noise model," but on line 143, they mention, "The proposed work introduces an Approximate Multiplication Noise (AMN) model." So, what is the relationship between the constant noise model and AMN? Does it mean that a constant noise model can be used to simulate the noise in the system when modeling approximate multiplication techniques?
4) Overall, the motivation behind this paper's research is not very clear. Starting from line 22, the author mentions that energy consumption can impact system scalability, but there is no mention or measurement of system scalability and reliability.
5) How would truncation multiplication operations introduce cumulative errors, and what would be their impact on system stability and reliability? How can algorithm optimization reduce the cumulative effects of truncation errors? How to select the appropriate truncation bits to enhance the resilience of truncation multiplication, thereby ensuring system stability even when affected by digital computation errors?
6) Since it is stated in line 41 that "Bit Error Rate (BER) and SER are key factors in ensuring the QOS," why doesn't this paper provide performance analysis specifically for BER?
Comments on the Quality of English LanguageModerate editing of English is required.
Reviewer 2 Report
Comments and Suggestions for Authors
1. Combine Introduction part with Towards energy efficient communication systems part and Contributions part or divide into Introduction and related work sections, and include one related work table.
2. Include one general diagram in the introduction part for better understanding of readers.
3. Add one paragraph with critical analysis to advocate why MF technique is employed for signal detection in the proposed work.
4. What is benefit of using QPSK, explain, why not different QAM or BPSK?
5. Add one table and add all simulations results and summarise the results at the end.
6. Reference the simulations parameters.
Comments on the Quality of English Language
Minor editing of English language required
Reviewer 3 Report
Comments and Suggestions for Authors
Comments on the Quality of English LanguageAuthor Response
Please see the attachment.

Round 2
Reviewer 1 Report
Comments and Suggestions for Authors
No further comment.
Reviewer 3 Report
Comments and Suggestions for Authors
I am satisfied with authors answers and approach to improving quality of the article. Thank you very much.